# Distinctive tasks of different cyanobacteria and associated bacteria in carbon as well as nitrogen fixation and cycling in a late stage Baltic Sea bloom

Falk Eigemann[1]*, Angela Vogts[1], Maren Voss[1], Luca Zoccarato[2], Heide Schulz-Vogt[1]

**1** Department of Biological Oceanography, Leibniz Institute for Baltic Sea Research Warnemünde, Rostock, Germany, **2** Department of Stratified Lakes, Leibniz-Institute for Freshwater Ecology and Inland Fisheries, Stechlin, Germany

* falkeigemann@gmail.com

**Data Availability Statement:** The forward and reverse reads from the Illumina sequencing were deposited at the European Nucleotide Archive

## Abstract

Cyanobacteria and associated heterotrophic bacteria hold key roles in carbon as well as nitrogen fixation and cycling in the Baltic Sea due to massive cyanobacterial blooms each summer. The species specific activities of different cyanobacterial species as well as the N- and C-exchange of associated heterotrophic bacteria in these processes, however, are widely unknown. Within one time series experiment we tested the cycling in a natural, late stage cyanobacterial bloom by adding $^{13}C$ bi-carbonate and $^{15}N_2$, and performed sampling after 10 min, 30 min, 1 h, 6 h and 24 h in order to determine the fixing species as well as the fate of the fixed carbon and nitrogen in the associations. Uptake of $^{15}N$ and $^{13}C$ isotopes by the most abundant cyanobacterial species as well as the most abundant associated heterotrophic bacterial groups was then analysed by NanoSIMS. Overall, the filamentous, heterocystous species *Dolichospermum* sp., *Nodularia* sp., and *Aphanizomenon* sp. revealed no or erratic uptake of carbon and nitrogen, indicating mostly inactive cells. In contrary, non-heterocystous *Pseudanabaena* sp. dominated the nitrogen and carbon fixation, with uptake rates up to $1.49 \pm 0.47$ nmol N $h^{-1}$ $l^{-1}$ and $2.55 \pm 0.91$ nmol C $h^{-1}$ $l^{-1}$. Associated heterotrophic bacteria dominated the subsequent nitrogen remineralization with uptake rates up to $1.2 \pm 1.93$ fmol N $h^{-1}$ cell $^{-1}$, but were also indicative for fixation of di-nitrogen.

## Introduction

Cyanobacterial blooms are a worldwide phenomenon in limnic, brackish and marine systems. In the Baltic Sea, blooms occur regularly during summer [1], and due to their high biomasses they significantly add to eutrophication [2,3]. Start of blooms is promoted by rising water temperatures and low N:P ratios after N-depletion due to the capability of atmospheric nitrogen fixation by several cyanobacterial species [1,3,4]. Total cyanobacterial nitrogen fixation in the Baltic Sea was estimated at 370 kt $yr^{-1}$ [2], and may contribute up to 55% of total nitrogen input [5,6]. Furthermore, filamentous cyanobacteria may contribute up to 44% of the

under the accession number PRJEB23316 (sample B15_3). All other data are in the paper and its Supporting Information files.

**Funding:** This work was supported by Human Frontiers Science Program (RGP0020/2016) to MV. This work was also supported by the SIMS instrument was funded by the German Federal Ministry of Education and Research (BMBF), grant identifier 03F0626A to AV. And this work was also supported by Leibniz Association to FE and HS-V. The funders had no role in study design, data collection and analysis, decision to publish, or preparation of the manuscript.

**Competing interests:** The authors have declared that no competing interests exist.

community primary production [7]. The major part of nitrogen and carbon fixation is performed in the early summer, followed by a peak in biomass, and ultimately the decay of the bloom in which predominantly recycling processes occur [6,8].

Cyanobacteria live in close associations with heterotrophic bacteria, and interactions between them may range from symbiosis to competition [9,10]. These interactions strongly influence carbon and nutrient cycling and thereby the stability of aquatic food webs [11,12]. In phytoplankton blooms, heterotrophic bacteria may provide macronutrients via recycling (or fixation) but may be also competitors for inorganic nutrients [11]. Especially at the late stages of cyanobacterial blooms, the associated heterotrophic bacteria may be responsible for a significant share of elemental cycling and fluxes, i.e. for the input of nutrients and organic matter in the ecosystem due to remineralization. Studies on the role of associated bacteria at these late cyanobacterial bloom stages, however, are lacking.

The predominant cyanobacterial genera in Baltic Sea blooms are *Aphanizomenon*, *Nodularia*, *Dolichospermum*, *Pseudanabaena* and *Synechococcus*, whereby the dominant groups and species may differ between years and stage of the bloom [12]. The first three mentioned genera are filamentous and heterocystous, and may form dense surface scums [1]. Baltic Sea *Synechococcus* sp. and *Pseudanabaena* sp. are supposed to be incapable of nitrogen fixation [13,14], even though nitrogenase genes occur in *Pseudanabaena* [14,15]. Thus, *Aphanizomenon*, *Nodularia*, and *Dolichospermum* are thought to dominate the biological nitrogen input into the Baltic Sea [14]. Recently, however, heterotrophic bacteria were shown to be capable of nitrogen fixation at depth in the central Baltic Sea [16] and may even be the principle $N_2$ fixing organism in a Baltic Sea estuary [17]. However, studies that examined carbon and nitrogen fixation in cyanobacterial blooms and associated heterotrophic bacteria mostly focussed on single cyanobacterial species [7,18], or neglected associated bacteria as well as the fate of the fixed carbon and nitrogen in the associations [14]. In the present study, we incubated a natural late stage Baltic Sea cyanobacterial bloom with $^{13}C$ bi-carbonate and $^{15}N_2$, and followed the uptake over time by means of NanoSIMS. Thereby, we aimed at unravel the specific contribution of different cyanobacterial species and associated heterotrophic bacteria in carbon and nitrogen fixation as well as the fate of the fixed carbon and nitrogen in the associations.

## Material and methods

### Incubation experiments

A natural cyanobacterial bloom was sampled at station TransA (58˚43.8‘N, 18˚01.9‘E, Fig 1) on 13.08.2015. Positive phototactic zooplankton was removed by means of a light trap and bloom samples were concentrated until a cyanobacterial chl. a concentration of 9 µg l$^{-1}$ was reached (measured with a PHYTO-PAM, Heinz Walz GmbH). At Askö laboratory (ca. 1 h transfer), five 176 mL opaque Nalgene bottles were filled with the concentrated bloom till overflowing and sealed with septum caps enabling addition and retrieval of liquids with syringes.

For $^{15}N$ addition, 1 mL of the sample was removed and subsequently 1 mL 99% pure $^{15}N_2$ gas injected with a syringe, resulting in 31.68 atom % $^{15}N$. For amending $^{13}C$, 5 ml sample were removed with a syringe and subsequently 5 ml F/2 medium [19] without nitrogen source, adjusted to 8 PSU and spiked with 0.4 g NaH$^{13}CO_3$ added (final concentration 108.36 atom % $^{13}C$). Incubation times were 10 min, 30 min, 1 h, 6 h and 24 h. Bottles were incubated in an incubation chamber at 16.5 ± 0.5˚C at approximately 60 µmol photons s$^{-1}$ m$^{-2}$ (delivered from ROHS 36W 840 light bulbs), resembling the natural conditions of sampling under constant light (S1 Table).

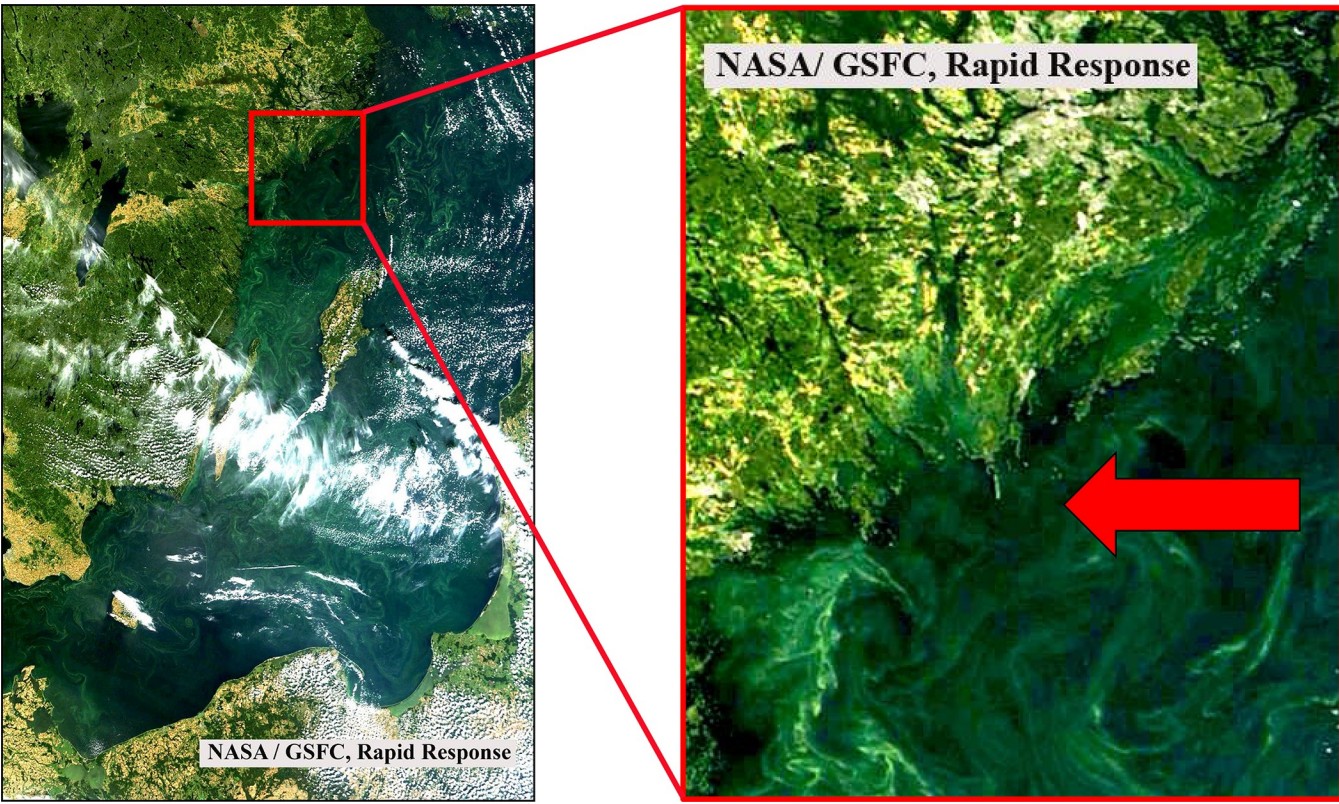

**Fig 1. True color satellite image of a cyanobacterial bloom in the Baltic Sea on August 13, 2015 derived from MODIS/Terra (NASA/GSFC, Rapid Response).** The arrow in the zoom image on the right side points towards the sampling station TransA.

## Sampling

Each sample was fixed with formaldehyde (2% final concentration) for 3 h in the dark at room temperature, and filtered gently onto 3 μm pore width (we only aimed at the directly attached heterotrophic bacterial fraction) polycarbonate filters for later inspection with CARD-FISH and NanoSIMS. Before start of the incubation, 80 mL of the stock sample were filtered onto 3 μm pore width polycarbonate filter for DNA extraction of the associated bacterial fraction. For phytoplankton counting, a 100 mL subsample was fixed with an acidic Lugol solution [20] and counted according to the Utermöhl technique. To determine biomass percentages, the carbon content (μg l$^{-1}$) of each species was calculated using the official PEG Biovolume Report 2016 (International Council for the Exploration of the Sea) for phytoplankton species and the carbon content per counting unit for the respective size class.

## DNA extraction

DNA was extracted as described in [21] with modifications. Briefly, the filters were cut into pieces and mixed with sterilized zirconium beads, 500 μl of phenol/chloroform mix, and 500 μl of SLS extraction buffer. After centrifugation of the mixture, the supernatant was transferred to another tube and the process was repeated. DNA was precipitated overnight at −20˚C. The pellet was washed with ethanol, dried, and resolved in autoclaved DEPC-treated water.

## PCR and sequencing

For PCRs, 10 ng of DNA was added to autoclaved DEPC-treated water, 10× PCR buffer, BSA, $MgCl_2$, dNTPs, forward and reverse primers, and native Taq polymerase. Bacterial DNA was amplified using the primers 341f and 805r [22], under the following conditions: 30 cycles of denaturation for 40 s at 95˚C, 40 s of annealing at 53˚C, and 1 min of elongation at 72˚C. PCR products were cleaned with the Nucleospin kit following the manufacturer's instructions and shipped to LGC Genomics GmbH (Berlin). Illumina MiSeq V3 sequencing with 300 bp paired-end reads was performed using the 16S primers 341F and 785R. The forward and reverse reads were deposited at the European Nucleotide Archive under the accession number PRJEB23316 (sample B15_3). Taxonomic identification of the associated bacterial community, was performed as described in [23] with the NGS analysis pipeline of the SILVA rRNA gene database project (SILVAngs 1.3).

## CARD-FISH analyses

The Illumina runs mostly yielded Alphaproteobacteria and Cytophaga/Bacteroidetes, and probes Alf968 [24] and CF968 [25] were chosen for analyses of associated heterotrophic bacteria. CARD FISH analyses were computed as described in [26] with modifications: Filter pieces were doused in 0.2% fluid agarose, dried, and subsequently incubated for 60 min at 37˚C in 10 mg ml$^{-1}$ lysozym solution and thereafter for 15 min at 37˚C with achromopeptidase (180 U ml$^{-1}$). For inactivation, filter pieces were doused subsequently to 1x PBS, autoclaved MilliQ and 99% ethanol and following placed for 10 min in 0.01 M HCl at room temperature. Hybridization with horseradish peroxidase labeled 16S rRNA probes Alf968 and CF968 were carried out at 35˚C with 55% formamide for 3.5 and 4 h, respectively. Signal amplification was achieved with Oregon green 488-X bound to tyramide as described in [27]. After hybridization, filter pieces were stained with 4,6-diamidin-2-phenylindol (DAPI) solution for unspecific counter-staining of all cells.

## Laser-Scanning Microscope, Scanning electron microscope and sputtering

Spots of interest were determined by fluorescence microscopy and subsequently laser marked with a laser microdissectional microscope. For confirmation of associated bacteria and cyanobacterial species, SEM analyses were performed. Therefore, filter pieces were covered with approximately 8 nm gold in a sputter coater (Cressington108 auto-sputter coater). Samples were analyzed with a Scanning electron microscope (Zeiss Merlin VP compact) with the Zeiss Smart SEM Software. Before NanoSIMS analyses, filter pieces were covered with ca. 30 nm additional gold with a sputter coater (see above).

## NanoSIMS measurements

SIMS imaging was performed using a NanoSIMS 50L instrument (Cameca, France). A $^{133}Cs^+$ primary ion beam was used to erode and ionize atoms of the sample. Among the received secondary ions, images of $^{12}C^-$, $^{13}C^-$, $^{12}C^{14}N^-$ and $^{12}C^{15}N^-$ were recorded simultaneously for cells at the laser microdissectional (LMD)-marked spots using electron multipliers as detectors. The mass resolving power was adjusted to suppress interferences at all masses allowing, e.g. the separation of $^{12}C^{15}N^-$ from interfering ions such as $^{13}C^{14}N^-$. Prior to the analysis, sample areas of 30×30 μm were sputtered for 2 min with 600 pA to erode the gold and reach the steady state of secondary ion formation. The primary ion beam current during the analysis was 1 pA; the scanning parameters were 512×512 px for areas of 20–30 μm, with a dwell time of 250 μs per pixel.

## Analyses of NanoSIMS measurements

All NanoSIMS measurements were analysed with the Matlab based program look@nanosims [28]. Briefly, the 60 measured planes were checked for inconsistencies and all usable planes accumulated, regions of interest (i.e. cells of cyanobacterial filaments, associated bacterial cells and filter regions without organic material for background measurements) defined based on $^{12}C^{14}N$ mass pictures, and $^{13}C/^{12}C$ as well as $^{15}N/^{14}N$ ratios calculated from the ion signals for each region of interest. Measurements of heterocysts in *Aphanizomenon* sp., *Dolichospermum* sp., and *Nodularia* sp. were avoided due to rapid transfer of fixed nitrogen. For analyses of each measurement, first the means of background measurements were determined, and this mean factorized for theoretical background values (0.11 for $^{13}C/^{12}C$ and 0.00367 for $^{15}N/^{14}N$). These factors were applied to all non-background regions of interest in the same measurement. For each time-point, values for each species (or bacterial group for the associated bacteria) were pooled (i.e. different cells in one measurement as well as different measurements) and means for each species (or bacterial group for the associated bacteria) for each time-point calculated. Work flow for an example spot from Card-FISH to NanoSIMS analyses is illustrated in Fig 2. The numbers of measured cells per species/group and time point, as well as overall measured areas per time point are given in S2 Table, the outcomes of all NanoSIMS analyses are given in S3 Table.

## Uptake rates of $^{13}C$ and $^{15}N$

Uptake rates for nitrogen and carbon were calculated as described in [29] according to the equation:

$$V(T^{-1}) = \left(\frac{1}{\Delta t}\right)\left(\frac{A_{PNf} - A_{PN0}}{A_{N2} - A_{PN0}}\right),$$

where $A_{N2}$ is the $^{15}N$ or $^{13}C$ enrichment of the N or C available for fixation; $A_{PN0}$ the $^{15}N$ or $^{13}C$ enrichment of particulate N or C at the start of the experiment; $A_{PNf}$, the $^{15}N$ or $^{13}C$ enrichment of particulate N or C at the end of the experiment; and

$$p(ML^{-3}T^{-1}) = \frac{V}{2} \times PX,$$

where PX is the concentration of N or C for the respective cyanobacterial species in the incubation bottles, or the cellular N or C content for the associated bacteria. The solubility of N and C was calculated using the Excel Sheet provided by Joe Montoya, based on [30] for $CO_2$ and [31] for $N_2$. For cyanobacteria gross uptake rates could be calculated per volume and time (absolute numbers were known). For the associated bacteria uptake rates were calculated per cell and time, because no absolute numbers of associated bacteria were existent. The C:N ratios in the cyanobacteria were assumed with 6.3 [14,32]. The size of the associated bacteria was assumed with 2 x1 (length x width) μm (SEM analyses), the carbon content of 0.35 pg C μm$^{-3}$ [33], and the C:N ratio of 5:1 [34]. We are aware that the used "bubble-method" for injection of $N_2$ gas assumes an instantaneous equilibrium between the $^{15}N_2$ gas bubble and the $N_2$ dissolved in water, which in fact may be time-delayed [35], and ultimately leads to an underestimation of fixation rates. Thus, especially at the early measuring points (10 and 30 min), the calculated rates should be considered as proxy values with percentage errors up to 70% [36].

## Data analyses

All data were analysed with R [37] and R studio [38]. To test for differences in stable isotope ratios between species/groups or between different time-points in the same group/species,

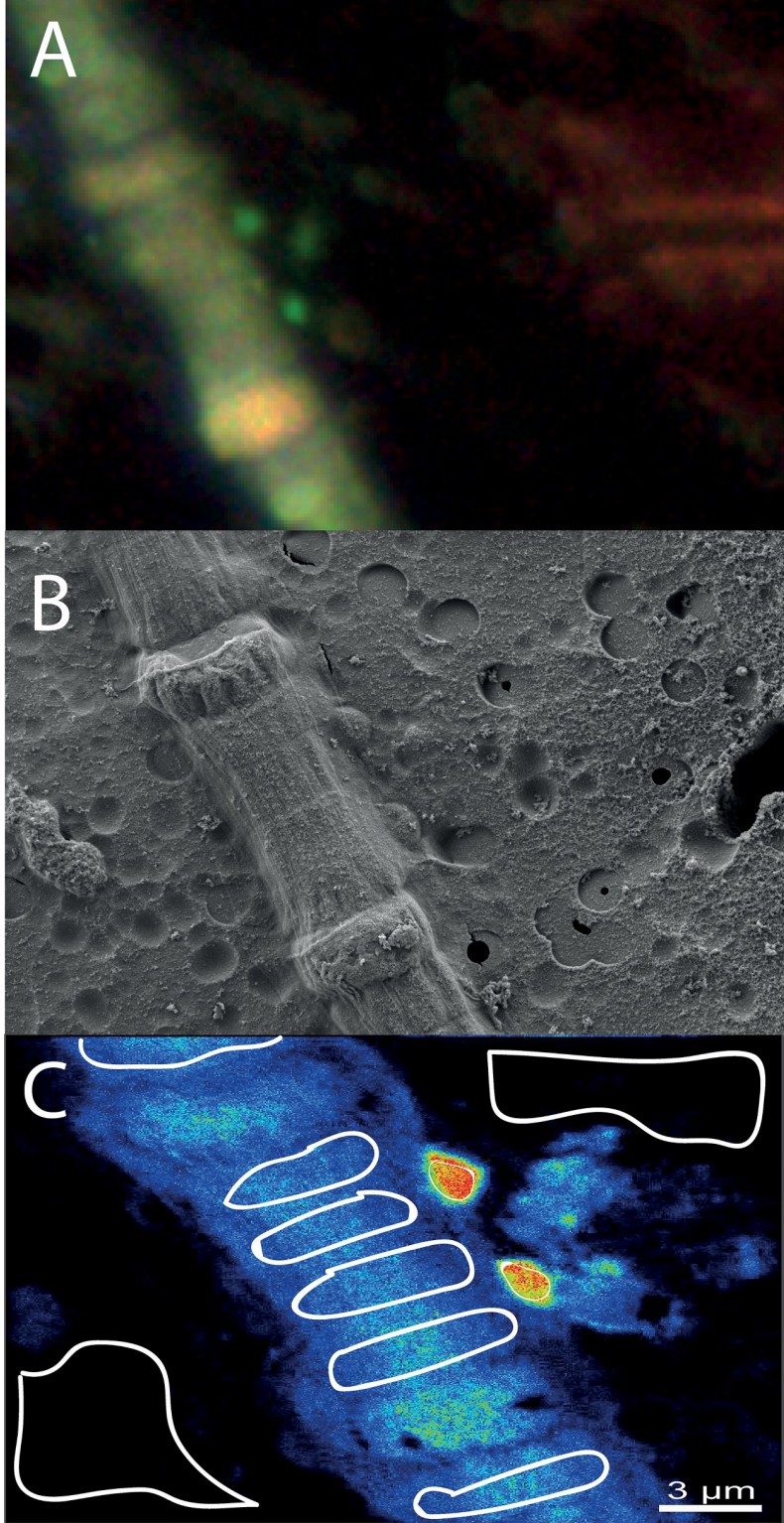

**Fig 2. Work flow for analyses of cyanobacteria and their associated heterotrophic bacteria.** A: Card-FISH image of a *Nodularia* sp. filament with two associated Alphaproteobacteria taken with a laser microdissectional microscope. The marking arrow can be seen at the right side. B: Scanning electron microscope image of the same spot for confirmation of associated bacteria (middle-right side of the filament) and identification of *Nodularia* sp. The tip of the marking arrow can be seen at the right side of the image. C: accumulated NanoSIMS images of the same spot with blue (low) to

red (high) $^{15}N$ signal (as example). The circled areas display the regions of interest, where $^{13}C/^{12}C$ and $^{15}N/^{14}N$ ratios were calculated. Control (filter without cyanobacteria or heterotrophic bacteria) regions can be seen at the top-right and down-left, *Nodularia* sp. regions are displayed in the bluish part, and the associated Alphaproteobacteria by the smaller regions in the reddish part of the image.

ANOVAS (analyses of variance) with subsequent Tukey HSD posthoc tests with the package agricolae [39] were performed. Likewise, the impact of the host species on the stable isotope uptake of the associated bacteria was tested with ANOVAs, by comparing associated bacterial cells from different hosts. Possible cell-to-cell transfer of $^{13}C$ and $^{15}N$ between host and associated bacteria were tested by calculating linear models of $^{13}C/^{12}C$ and $^{15}N/^{14}N$ ratios between the host cells and the associated bacterial cells for each incubation period. To test for correlations between $^{13}C$ and $^{15}N$ uptake, linear models were calculated with the lm function. To test for differences in relations of $^{13}C$ to $^{15}N$ uptake between species/groups, dissimilarity matrices (horn distances) were calculated with a xy (x = $^{13}C/^{12}C$, y = $^{15}N/^{14}N$) system, and subsequently ANOSIM analyses performed with the vegan package [40]. To test for differences in $^{13}C/^{15}N$ uptake relations between functional groups, ANCOVAs with and without interactions between the factor and the co-variable were calculated with linear models. Here, $^{13}C$ uptake was set as dependent variable, $^{15}N$ uptake as co-variable, and the functional group as factor. Next, ANOVAs were calculated for both ANCOVAs to test for differences in the slopes of the linear models.

## Results

### Community composition of the phytoplankton and associated bacteria

The phytoplankton community was dominated by the cyanobacteria *Aphanizomenon* sp. (33% biomass), *Nodularia* sp. (30% biomass), *Pseudanabaena* sp. (9% biomass) and *Dolichospermum* sp. (8% biomass), which together accounted for 80% of the total biomass (Fig 3A). The most abundant associated bacteria belonged to Alphaproteobacteria (39%), Cytophaga/Bacteroidetes (20%), Gammaproteobacteria (18%), Verrucomicrobia (6%), Planctomycetes (5%), Betaproteobacteria (4%) and Actinobacteria (1%, Fig 3B).

The general appearance of the bloom (Fig 4A), and microscopy of cyanobacteria (Fig 4B–4E) both indicated a late stage of the bloom (especially the "curly" appearance of *Nodularia* sp.), with many associated bacteria to the heterocystous species (Fig 4F).

### Bi-carbonate uptake of cyanobacteria and associated heterotrophic bacteria

Significant differences in the $^{13}C$ incorporation between the bacterial groups were observed at all sampling points (Fig 5). *Pseudanabaena* sp. showed the highest $^{13}C/^{12}C$ ratios at all sampling points with continuously increasing incorporation of $^{13}C$ over time. At the early time points (10, 30 and 60 min), all other species/groups displayed a $^{13}C/^{12}C$ ratio close to the natural occurring value of 0.011 (Fig 5). After 6 and 24 h of incubation, however, Cytophaga/Bacteroidetes revealed the second highest $^{13}C/^{12}C$ ratios, corresponding to significant $^{13}C$ enhancements with a more than two- and ten-fold increase of the natural occurring ratio after 6 and 24 h, respectively (Fig 5). Mentionable, the filamentous cyanobacteria *Aphanizomenon* sp., *Dolichospermum* sp. and *Nodularia* sp. did not display elevated $^{13}C/^{12}C$ ratios over the whole 24 h incubation period with two exceptions: *Aphanizomenon* sp. revealed enhanced ratios after 6 h and *Dolichospermum* sp. after 24 h of incubation (Fig 5, S4 Table). To test for a possible impact of the host-species on $^{13}C$ uptake of the associated bacteria, we compared the $^{13}C/^{12}C$ ratios obtained from Alphaproteo- and Cytophaga/Bacteroidetes bacteria from

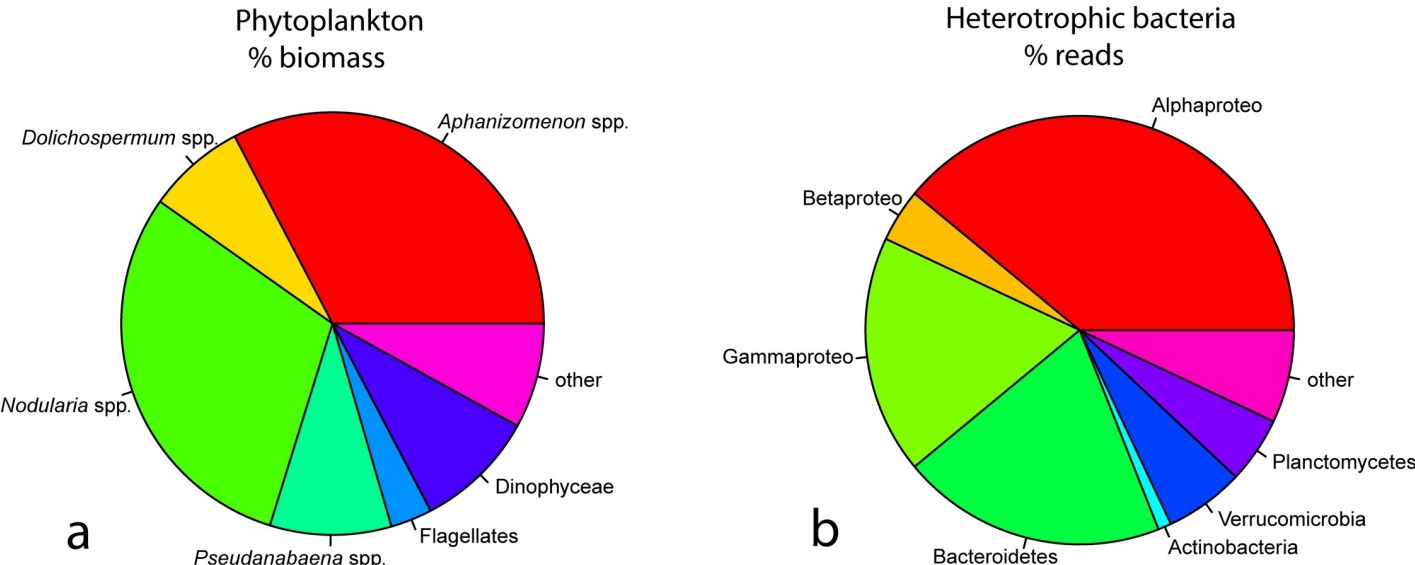

**Fig 3.** A: Pie chart for the most abundant phytoplankton groups (left side, in % biomass). B: Pie chart for the most abundant bacterial groups (right side, in % of sequencing reads).

different host species. In most cases, however, no significant differences occurred between the hosts (S4 Table). Especially in the 6 and 24 h exposures, where increased $^{13}C/^{12}C$ ratios were obtained for both of the associated bacterial groups (Fig 5), no impact of the host species could be seen (S4 Table). Linear models on $^{13}C$ uptake between the host cells and the associated bacterial cells did not suggest cell-cell transfer of $^{13}C$ except for the 60 min incubation ($R^2$ = -0.05,

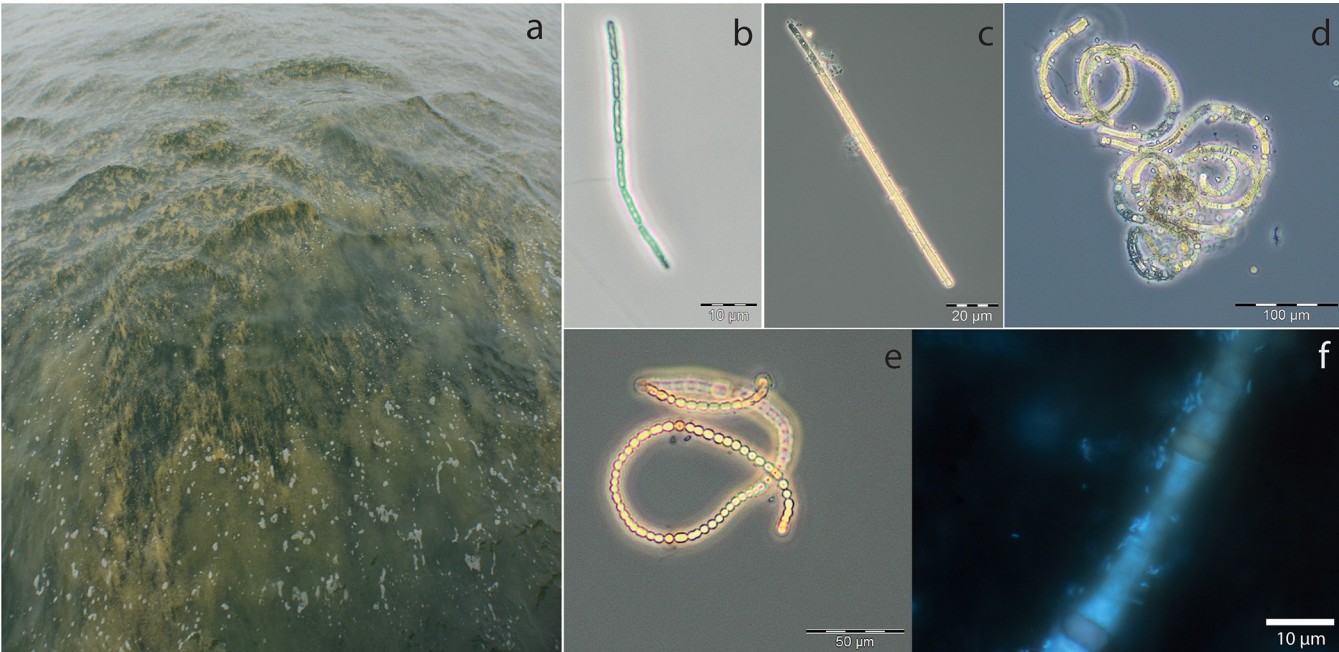

**Fig 4.** Appearance of the bloom at the day of sampling (a), and microscopic images of *Pseudanabaena* sp. (b), *Aphanizomenon* sp. (c), *Nodularia* sp. (d), *Dolichospermum* sp. (e), and a DAPI stained sample with *Nodularia* sp. and associated bacteria.

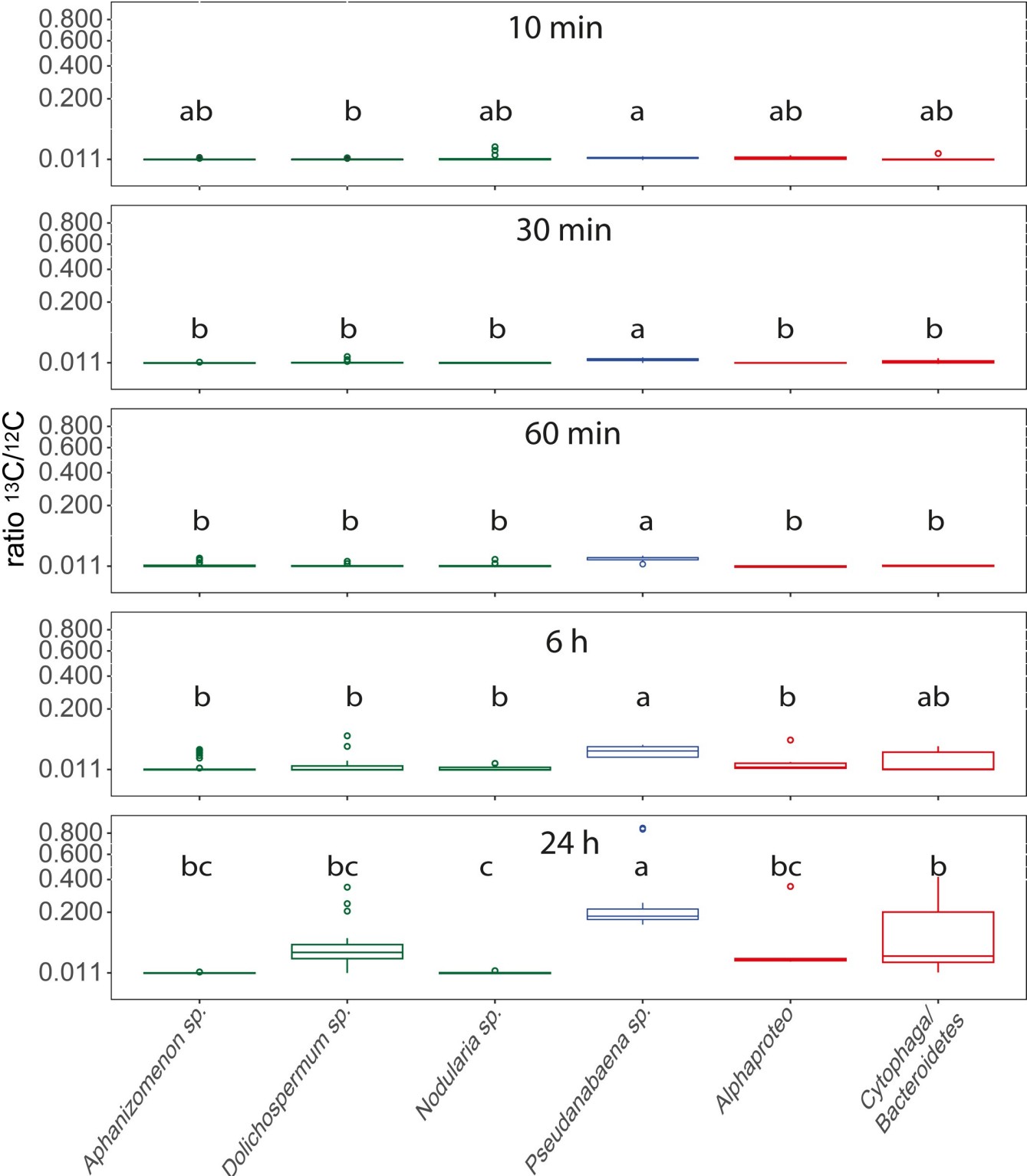

**Fig 5. Boxplots of $^{13}C/^{12}C$ ratios for *Aphanizomenon* sp., *Dolichospermum* sp., *Nodularia* sp., *Pseudanabaena* sp., Alphaproteobacteria and Cytophaga/ Bacteroidetes bacteria over time, with square root transformed y axis.** Values originate from pooled data for the respective species from different measurements and cells (S2 Table). Lower case letters above the boxplots refer to different groups of Tukey HSD Post-Hoc tests. Heterocystous cyanobacteria are displayed in green, non-heterocystous cyanobacteria in blue, and associated heterotrophic bacteria in red.

**Table 1. Carbon and nitrogen uptake rates ± standard deviation given in nmol C or N h⁻¹ l⁻¹ for cyanobacteria, and fmol C or N h⁻¹ cell⁻¹ for associated bacteria.**

|  | 10 min | 30 min | 60 min | 6 h | 24 h |
|---|---|---|---|---|---|
|  | $^{13}C$ uptake nmol C h$^{-1}$ l$^{-1}$ (fmol C h$^{-1}$ cell$^{-1}$ for associated bacteria) | | | | |
| *Aphanizomenon sp.* | 0.3 ± 3.52 | 0.00 ± 0.57 | 1.06 ± 2.69 | 0.41 ± 1.35 | 0.00 ± 0.02 |
| *Dolichospermum sp.* | 0.06 ± 0.76 | 0.25 ± 0.71 | 0.05 ± 0.2 | 0.07 ± 0.13 | 0.27 ± 0.18 |
| *Nodularia sp.* | 5.88 ± 20.28 | 0.00 ± 0.95 | 0.4 ± 1.5 | 0.19 ± 0.4 | 0.00 ± 0.03 |
| *Pseudanabaena sp.* | 2.48 ± 1.5 | 1.98 ± 0.89 | 2.55 ± 0.91 | 1.13 ± 0.72 | 1.87 ± 1.08 |
| *Alphaproteo** | 0.59 ± 0.9 | 0.00 ± 0.04 | 0.00 ± 0.07 | 0.13 ± 0.23 | 0.2 ± 0.32 |
| *Cytophaga/Bacteroidetes** | 0.12 ± 0.09 | 0.24 ± 0.45 | 0.03 ± 0.06 | 0.19 ± 0.27 | 0.31 ± 0.34 |
|  | $^{15}N$ uptake nmol N h$^{-1}$ l$^{-1}$ (fmol N h$^{-1}$ cell$^{-1}$ for associated bacteria) | | | | |
| *Aphanizomenon sp.* | 1.03 ± 1.6 | 0.00 ± 1.27 | 0.28 ± 0.42 | 0.17 ± 0.4 | 0.00 ± 0.02 |
| *Dolichospermum sp.* | 0.73 ± 0.45 | 0.17 ± 0.3 | 0.05 ± 0.08 | 0.09 ± 0.17 | 0.04 ± 0.03 |
| *Nodularia sp.* | 8.07 ± 18.4 | 0.03 ± 0.87 | 0.18 ± 0.53 | 0.19 ± 0.23 | 0.00 ± 0.02 |
| *Pseudanabaena sp.* | 1.49 ± 0,47 | 0.8 ± 0.56 | 0.84 ± 0.17 | 0.48 ± 0.31 | 0.17 ± 0.05 |
| *Alphaproteo** | 0.00 ± 0.08 | 0.2 ± 0.73 | 0.31 ± 0.76 | 1.15 ± 1.29 | 0.34 ± 0.2 |
| *Cytophaga/Bacteroidetes** | 0.95 ± 1.01 | 0.36 ± 0.53 | 1.2 ± 1.93 | 0.67 ± 0.92 | 0.25 ± 0.17 |

Lines with rates of the associated bacteria are indicated with an asterisk.

0.12, 0.24, -0.03, -0.05; p = 0.84, 0.24, 0.01, 0.48, 0.75, for 10 min, 30 min, 60 min, 6 h and 24 h incubation, respectively). The calculated uptake rates of the cyanobacteria were highest for *Pseudanabaena* sp. after 60 min with 2.55 ± 0.91 nmol C h⁻¹ l⁻¹, and from the associated bacteria for Cytophaga/Bacteroidetes bacteria with 0.31 ± 0.34 fmol C h⁻¹ cell⁻¹ after 24 h of incubation (Table 1).

## $^{15}N_2$ uptake of cyanobacteria and associated heterotrophic bacteria

For all time points, significant differences of $^{15}N$ incorporation between the species/groups occurred (Fig 6). After 30 min *Pseudanabaena* sp. (which reveals the highest $^{15}N$ incorporation), and the associated heterotrophic bacteria showed enhanced $^{15}N/^{14}N$ ratios (Fig 6). Between 1 and 6 h of incubation especially the Alphaproteobacteria increased their $^{15}N/^{14}N$ ratios, and after 24 h of incubation pronounced differences between the species occurred, with associated Alphaproteobacteria showing the highest $^{15}N$ incorporation (mean = 0.0143 ± 0.0059, almost 4-times increased $^{15}N/^{14}N$ ratios compared to the natural ratio). In general, after 24 h of incubation the associated bacteria revealed the highest ratios, followed by *Pseudanabaena* sp., whereas the heterocystous cyanobacteria displayed even after 24 h of incubation $^{15}N/^{14}N$ ratios close to the natural value (Fig 6).

Comparisons of the $^{15}N/^{14}N$ ratios in each species/group between different incubation times revealed significant $^{15}N$ incorporation in most species/groups, but inconsistent $^{15}N$ uptake in the heterocystous species (Fig 6). In general, the heterocystous cyanobacteria do not display pronounced $^{15}N_2$ uptake over time. In contrast, *Pseudanabaena* sp. displays significantly enhanced $^{15}N/^{14}N$ ratios from 6 h of incubation onwards, with steadily increasing values over time and significant differences also between 6 and 24 h of incubation (Fig 6, S4 Table). Also Alphaproteobacteria and Cytophaga/Bacteroidetes reveal steadily increasing $^{15}N/^{14}N$ ratios over the 24 h incubation period, with significant differences between almost all incubation times (Fig 6, S4 Table). The separation of the obtained $^{15}N/^{14}N$ values of associated Alphaproteo- and Cytophaga/Bacteroidetes bacteria by the host species did not reveal differences between the host species (S4 Table). Linear models between the $^{15}N/^{14}N$ ratios of heterocystous cyanobacterial cells that carry associated bacteria and the associated bacteria did not

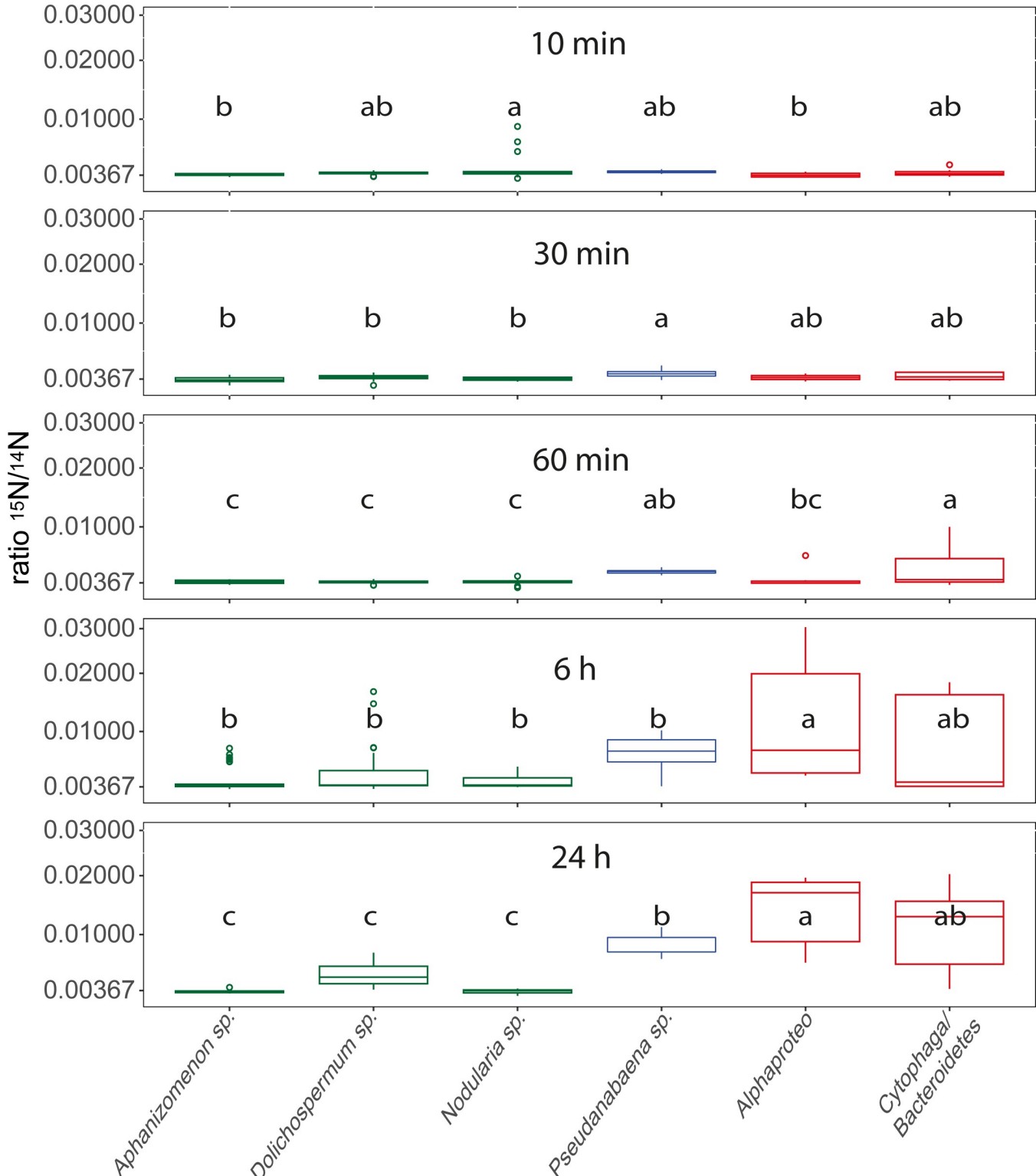

**Fig 6. Boxplots of $^{15}$N/$^{14}$N ratios for *Aphanizomenon* sp., *Dolichospermum* sp., *Nodularia* sp., *Pseudanabaena* sp., Alphaproteobacteria and Cytophaga/ Bacteroidetes bacteria over time with square root transformed y-axis.** Values originate from pooled data for the respective species from different spots and cells (S2 Table). Lower case letters above the boxplots refer to different groups of Tukey HSD Post-Hoc tests. Heterocystous cyanobacteria are displayed in green, non-heterocystous cyanobacteria in blue, and associated heterotrophic bacteria in red.

suggest dependencies of $^{15}$N uptake between the host and the associated bacterium, with the exception of 30 min incubation ($R^2$ = -0.02, 0.64, 0.08, -0.03, 0.1; p = 0.48, 0.02, 0.1, 0.39, 0.1 for 10 min, 30 min, 1h, 6 h, and 24 h incubation, respectively). The uptake rates were highest for *Nodularia* sp. with 8.07 ± 18.4 nmol N h$^{-1}$ l$^{-1}$ after 10 min of incubation. However, if excluding the 10 min incubation due to experimental uncertainties, *Pseudanabaena* sp. revealed the highest incorporation rates with 0.84 ± 0.17 nmol N h$^{-1}$ l$^{-1}$ after 1 h of incubation. For the associated bacteria Cytophaga/Bacteroidetes displayed the highest incorporation of $^{15}$N with 1.2 ± 1.93 fmol N h$^{-1}$ cell$^{-1}$ after 1 h incubation (Table 1).

## Species- and group specific relations of $^{13}$C to $^{15}$N uptake

Significant differences between the species/bacterial groups occurred for all time points for relations of $^{13}$C against $^{15}$N uptake (ANOSIM, each p = 0.001), although different R values were obtained for different exposure times (R = 0.2387, 0.4203, 0.3098, 0.215, 0.585, for 10, 30, 60 min, 6 and 24 h exposure, respectively), indicating most pronounced differences in the relation of $^{13}$C to $^{15}$N uptake between the species/groups after 24 h of incubation. In general, *Pseudanabaena* sp. was the most noticeable species in the $^{13}$C uptake (starting with the 30 min exposure), and the associated Alphaproteo and Cytophaga/Bacteroidetes bacteria in the $^{15}$N uptake (starting after 60 min of exposure, Fig 7). The heterocystous cyanobacteria revealed a high patchiness with few cells displaying prominent $^{13}$C uptake (Fig 5), but mostly did not show obvious uptake of either $^{13}$C or $^{15}$N (Fig 7, Table 1). Pooling the different species (for bacteria groups) into the functional groups heterocystous cyanobacteria (*Aphanizomenon* sp., *Dolichospermum* sp., and *Nodularia* sp.), non-heterocystous cyanobacteria (*Pseudanabaena* sp.), and associated bacteria (Alphaproteo- and Cytophaga/Bacteroidetes bacteria), and plotting of the $^{13}$C/$^{12}$C and $^{15}$N/$^{14}$N ratios against time, revealed specific tasks of the functional groups (Fig 7, Table 2). The associated bacteria predominantly display enhanced $^{15}$N/$^{14}$N ratios, with the highest ratios after 6 h incubation, whereas non-heterocystous cyanobacteria reveal the highest $^{13}$C/$^{12}$C ratios with a time dependent increase. Controversially, only few heterocystous cyanobacteria show increased $^{13}$C/$^{12}$C and/or $^{15}$N/$^{14}$N ratios (Fig 7).

Group specific behavior was corroborated by significantly different slopes between the functional groups in regression analyses of the $^{13}$C over $^{15}$N uptake for the different exposure times, despite the fact that significant correlations between $^{13}$C and $^{15}$N uptake occurred for all groups (Table 2). From 60 min of exposure onwards, the slopes of the associated bacteria are by far the steepest, corresponding to a predominant incorporation of $^{15}$N, whereas non-heterocystous cyanobacteria reveal flat slopes accompanying predominant incorporation of $^{13}$C (Table 2).

## Discussion

The present study determined the specific contribution of four different cyanobacterial species and the two most abundant associated bacterial groups in carbon as well as nitrogen fixation and cycling in late stage cyanobacterial bloom associations. Altogether, the cyanobacterium *Pseudanabaena* spp. dominated the carbon assimilation as well as nitrogen fixation at the early time-points, and the associated Alphaproteo- and Cytophaga/Bacteroidetes bacteria the nitrogen cycling and possibly N$_2$ fixation at the later time-points. The filamentous, heterocystous cyanobacteria *Nodularia* sp., *Dolichospermum* sp., and *Aphanizomenon* sp. on the other hand, either showed no or erratic carbon and nitrogen uptake. Among the associated heterotrophic bacteria Cytophaga/Bacteroidetes were more active in the carbon cycling, whereas Alphaproteobacteria revealed higher activity in nitrogen cycling. However, high intra-species variability

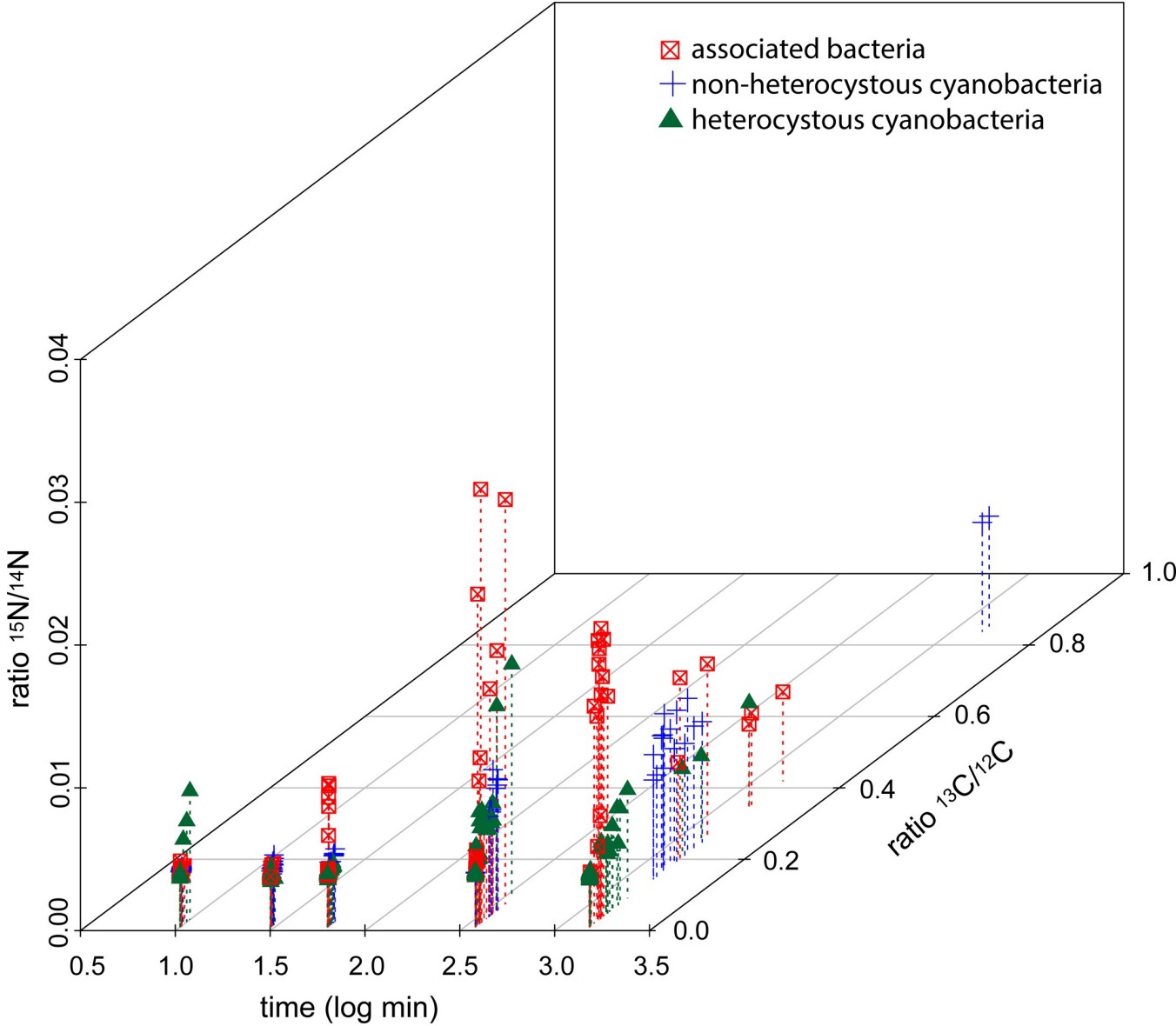

**Fig 7.** $^{13}$C/$^{12}$C (z axis) and $^{15}$N/$^{14}$N (y axis) ratios plotted against the exposure time (log transformed x axis) for the different functional groups (heterocystous cyanobacteria, non-heterocystous cyanobacteria, associated bacteria). The color and symbol legend is given directly in the figure.

was observed in all examined species, which partly impeded significant differences in isotope uptake between species and time points.

## Bi-carbonate uptake of cyanobacteria and associated heterotrophic bacteria

Surprisingly, *Pseudanabaena* sp. and not the heterocystous cyanobacteria was the most prominent species in carbon assimilation (Fig 5), with fixation rates up to 2.55 nmol C h$^{-1}$ l$^{-1}$. Indeed, carbon fixation rates of *Pseudanabaena* sp. were much higher than the rates for the heterocystous species *Aphanizomenon* sp., *Dolichospermum* sp., and *Nodularia* sp. together (Table 1, exception after 10 min of incubation due to 3 extraordinary high measurements in *Nodularia* sp.). However, in combined measurements of June, July and August in the preceding seasons

**Table 2. Regression analyses of [13]C over [15]N uptake for the functional groups heterocystous cyanobacteria (*Aphanizomenon* sp., *Dolichospermum* sp., *Nodularia* sp.), associated bacteria (Alphaproteo and Cytophaga/Bacteroidetes bacteria), and non-heterocystous cyanobacteria (*Pseudanabaena* sp.) for the different incubation times.**

| Incubation time | 10 min | 30 min | 60 min | 6 h | 24 h |
|---|---|---|---|---|---|
| Heterocystous cyanobacteria | Y = 0.001+0.246x, $R^2$ = 0.88, p = 0.000 | Y = 0.004–0.024x, $R^2$ = 0.00, p = 0.308 | Y = 0.001+0.03x, $R^2$ = 0.16, p = 0.000 | Y = 0.002+0.149x, $R^2$ = 0.82, p = 0.000 | Y = 0.00+0.01x, $R^2$ = 0.56, p = 0.000 |
| Associated bacteria | Y = 0.004+0.018x, $R^2$ = 0.02, p = 0.471 | Y = 0.003+0.105x, $R^2$ = 0.45, p = 0.06 | Y = -0.02+1.94x, $R^2$ = 0.52, p = 0.000 | Y = 0.004+0.329x, $R^2$ = 0.4, p = 0.007 | Y = 0.015–0.022x, $R^2$ = 0.24, p = 0.02 |
| Non-heterocystous cyanobacteria | Y = 0.004+0.028x, $R^2$ = 0.04, p = 0.169 | Y = 0.003+0.105x, $R^2$ = 0.28, p = 0.02 | Y = 0.004+0.05x, $R^2$ = 0.72, p = 0.001 | Y = 0.003+0.117x, $R^2$ = 0.82, p = 0.000 | Y = 0.009–0.002x, $R^2$ = 0.02, p = 0.27 |
| Anova | F = 2.77, p = 0.066 | F = 5.69, p = 0.001 | F = 12.9, p = 0.000 | F = 60.42, p = 0.000 | F = 9.79, p = 0.000 |

Anova results display comparisons of regression slopes of the different functional groups (ANCOVAs with and without interactions between the factor (functional group) and the co-variable ([15]N uptake) were calculated with linear models, with [13]C uptake set as dependent variable. ANOVAs were then calculated between both ANCOVAs to test for differences in the regression slopes).

2012 and 2013, the three heterocystous species together accounted for ca. 5–250 nmol C h$^{-1}$ l$^{-1}$ [14]. Thus, the heterocystous cyanobacteria still hold key roles in carbon fixation in the Baltic Sea [14,41], with much higher fixation rates compared to the estimated ones of *Pseudanabaena* sp. in the present study. In our case, the appearance of the bloom and the curly phenotype of *Nodularia* sp. suggested a late stage of the bloom (Fig 4), and the low activity of *Aphanizomenon* sp., *Dolichospermum* sp., and *Nodularia* sp. cells might be attributed to inactive cells at the late bloom stage. *Pseudanabaena* sp. was still active and may be adapted to this situation where phosphorus supply by degrading blooms may be granted.

Measurements of heterotrophic bacteria at the later incubation times also revealed enhanced [13]C/[12]C ratios (Fig 5), and heterotrophic bacteria may also incorporate bi-carbonate [42]. However, the [13]C signal in heterotrophic bacteria arises after 6 h incubation which may be related to recycled organic carbon released by *Pseudanabaena* sp. and other cells. The higher proportion of Cytophaga/Bacteroidetes bacteria in the incorporation of [13]C compared to Alphaproteobacteria (Fig 5) fits the current knowledge on their ecology: Marine Cytophaga/Bacteroidetes are specialized in the degradation of high molecular weight compounds [43–45], which are especially exuded in high quantities in late stage and senescent blooms [46–48]. Alphaproteobacteria on the other hand preferentially use low molecular weight compounds such as amino acids [44] and may act complementary to Bacteroidetes/Cytophaga in cyanobacterial bloom associations [43]. Thus, the higher [13]C incorporation in Cytophaga/Bacteroidetes bacteria may display the recycling of complex organic material whereas the lower signal in the Alphaproteobacteria account for the incorporation of low molecular weight exudates.

## [15]N$_2$ uptake of cyanobacteria and associated heterotrophic bacteria

*Pseudanabaena* sp. showed [15]N$_2$ incorporation after 30 min of incubation, and was the only species with significantly increased [15]N/[14]N ratios at this time. Further, it was the species with the highest [15]N/[14]N ratios after 60 min of incubation (Fig 6). Until now, the non-heterocystous *Pseudanabaena* sp. was not shown to be involved in fixation of atmospheric nitrogen in the Baltic Sea [13,14], despite the presence of nitrogenase genes [15]. However, picocyanobacteria and non-heterocystous filamentous species were suspected for nitrogen fixation under specific conditions before [14]. Taking into account that *Pseudanabaena* sp. was the only species with increased [15]N/[14]N ratios at the early sampling points, our data suggest an active N$_2$ fixation by

*Pseudanabaena* sp., with fixation rates between 0.17 and 1.49 nmol N h$^{-1}$ l$^{-1}$ (Table 1). Thus, at this late stage of the bloom, *Pseudanabaena* sp. might have been responsible for the input of reactive nitrogen in the multi-species associations and ultimately into the nitrogen cycle of the Baltic Sea. Indeed, if converted to per cell rates, nitrogen fixation of *Pseudanabaena* sp. appears low with up to 0.07 fmol N cell $^{-1}$ h$^{-1}$ if compared to the heterocystous species *Nodularia spumigena* (11 fmol N cell$^{-1}$ h$^{-1}$, [18]) and *Aphanizomenon* sp. (1–11 fmol N cell$^{-1}$ h$^{-1}$, [7]). However, this difference might be attributed to the much smaller cell size of *Pseudanabaena* sp., and compensated by higher cell numbers. In a comparable study of a Baltic Sea cyanobacterial bloom, cumulative fixation rates for combined measurements of June, July and August of the heterocystous species *Dolichospermum* sp., *Nodularia* sp., and *Aphanizomenon* sp. were determined with ca. 0.5–80 nmol N l$^{-1}$ h$^{-1}$ [14], i.e. approximately one order of magnitude above that of *Pseudanabaena* sp. alone in the present study. Likewise to the carbon fixation, the inexistent nitrogen fixation of the heterocystous species in our study may be attributed to different stages of the blooms, with most cells of heterocystous species being inactive at the late stage of the bloom (Figs 5 and 6). Consistent with these results, early/mid-summer nitrogen fixation rates in the Baltic Sea were up to 30 times higher compared to late summer [8]. Thus, heterocystous cyanobacteria may still be the prime nitrogen fixers in the Baltic Sea [5,6], but the possible participation of *Pseudanabaena* spp. should not be neglected. If this temporal divided nitrogen fixation between different cyanobacterial species represents a general feature for the Baltic Sea needs to be investigated in consecutive studies.

The overall highest $^{15}$N/$^{14}$N ratios by the associated bacteria after 6 and 24 h of exposure are surprising, taking the high abundance of diazotrophic cyanobacteria and the low $^{15}$N incorporation of the hosts into account. Indeed, one would expect the converse role allocation, where associated heterotrophic bacteria reveal lower $^{15}$N/$^{14}$N ratios than their diazotrophic hosts [18,49]. However, our high $^{15}$N/$^{14}$N ratios were obtained after 6 and 24 h of incubation, and thus, similar to the $^{13}$C incorporation, the associated bacteria may have used recycled nitrogen that was originally fixed by cyanobacteria. Supporting this assumption, heterotrophic microorganisms in cyanobacterial associations dominated by *Aphanizomenon* sp. relied on recycled nitrogen [49], and *Aphanizomenon* sp. was shown to release up to 35% of the fixed nitrogen as $NH_4^+$ [7]. However, direct cell to cell transmission between hosts and associated bacteria was not indicated (see 3.2 and the linear models), and release and transfer of newly fixed $N_2$ was not indicated at a similar experiment during 12 h of incubation [14].

The role of heterotrophic bacteria in nitrogen fixation budgets for aquatic ecosystem were recently brought into focus [17,50], and might have been underestimated in preceding studies [51–53]. As examples, heterotrophic organisms dominated the nitrogen fixation in the South Pacific Gyre [52], and were also the principle nitrogen fixing organisms in a Baltic Sea estuary [17]. Indeed, there are hints that the associated bacteria in our study also performed nitrogen fixation themselves and not only used nitrogen released from other cells: First, if the associated bacteria would only recycle nitrogen that was fixed by other organisms, one would expect a dilution in the $^{15}$N/$^{14}$N ratios from the primary fixer to the secondary user [8,14,49], which is not the case (Figs 6 and 7). Second, already after 30 min heterotrophic bacterial cells possessed the overall highest $^{15}$N/$^{14}$N ratios (Fig 7), and this fast incorporation indicates active nitrogen fixation. Third, many Alphaproteobacteria [16,17,53] and Cytophaga/Bacteroidetes bacteria [54] possess nitrogenase genes, and are capable of nitrogen fixation. To validate heterotrophic nitrogen fixation we performed a gene functional analysis with the 16S data of the associated bacteria using paprica—PAthway PRediction by phylogenetIC plAcement [55]. In this analysis, however, only 1.2% of the associated bacteria yielded the full pathway (via ferredoxin) for nitrogen fixation (S5 Table) which does not support our assumption. Nevertheless, ecosystem key functions may be performed by low abundant bacteria [56,57], and the per cell fixation

rates of the associated bacteria were more than one dimension higher compared to *Pseudanabaena* sp. (1.2 vs 0.07 fmol N h$^{-1}$ cell$^{-1}$), and in the same dimension as uptake rates for the much bigger heterocystous cyanobacteria (0.1–32.7 fmol N cell$^{-1}$ h$^{-1}$) in the Baltic Sea [14]. Thus, given the high abundances of associated bacteria, heterotrophic nitrogen fixation might contribute significantly to bulk fixation at this late stage bloom. At this stage of the bloom, senescent phytoplankton exhibit high exudation and leaking rates (e.g. [46]), and create an environment with high levels of labile DOC that fuels heterotrophic nitrogen fixation [51,58,59]. This is corroborated with the linear models, where bacteria associated to inactive, senescent hosts showed the highest $^{15}$N uptake (S6 Table). However, until now prerequisites and regulation of heterotrophic nitrogen fixation as well as principle contradictions as fixation in oxygenated waters and at high nitrate and ammonium concentrations are poorly understood [51], and should move into the focus of upcoming studies.

## Relation of $^{13}$C to $^{15}$N uptake

Significant correlations between $^{13}$C and $^{15}$N uptake occurred in most species and at most time points (Table 2), which is in accordance with similar studies from cyanobacterial blooms in the Baltic Sea (e.g. [14]). Nevertheless, relations between carbon and nitrogen uptake indicated specific tasks of functional groups (Fig 7, Table 2). *Pseudanabaena* sp. (non-heterocystous cyanobacterium) clearly dominated the $^{13}$C uptake (Fig 5) throughout the whole incubation period, but was also the first species with increased $^{15}$N signals (Fig 6). For the $^{15}$N/$^{14}$N ratios, however, *Pseudanabaena* sp. was outpaced by the associated bacteria from 6 h incubation onwards (Fig 6), and revealed much lower per cell fixation rates (see above). Thus, the associated bacteria may have dominated the nitrogen cycling and possibly fixation at the later sampling points. This specification of functional groups was corroborated by significant different slopes in linear models calculated for correlations between $^{13}$C and $^{15}$N uptake (Table 2). The formation of distinct functional groups by different species in late stage bloom associations may ultimately result in the allocation of desired metabolic pathways to every member in the association, including members unable to perform these tasks [60,61]. The concerted action of diverse ecological functions by different functional groups was also proposed for a chlorophyte and its prokaryotic epiflora [62], and might be a general feature of multi-species associations.

## Supporting information

**S1 Table. Sampling conditions at station TransA.** Abiotic variables at the day of sampling at station TransA.
(DOCX)

**S2 Table. Regions of interest (ROIs) analysed.** A: Numbers of analysed cells per species/ group as well as total analysed areas (which may contain different species/bacterial groups) for each incubation time. B: Numbers of analysed associated bacterial cells with the respective cyanobacterial host species for each incubation time.
(DOCX)

**S3 Table. NanoSIMS analyses.** NanoSIMS analyses of $^{13}$C/$^{12}$C and $^{15}$N/$^{14}$N ratios for all measuring points and regions of interest (ROIs).
(XLSX)

**S4 Table. Statistics for $^{13}$C and $^{15}$N uptake.** ANOVAs and Post-Hoc analyses for differences in $^{13}$C and $^{15}$N uptake for the different species over time as well as differences in $^{13}$C and $^{15}$N uptake by the associated bacteria caused by different hosts.
(DOCX)

**S5 Table. Paprica—PAthway PRediction by phylogenetIC plAcement analyses.** Outcome of the gene functional analysis with the 16S data of the associated bacteria using paprica—PAthway PRediction by phylogenetIC plAcement [55].
(XLSX)

**S6 Table. Stable isotope rations for hosts and associated bacteria.** $^{13}C/^{12}C$ and $^{15}N/^{14}N$ ratios for host cells and associated bacterial cells.
(XLSX)

## Acknowledgments

We thank Annett Grütmüller for NanoSIMS measurements, the field station Askö laboratory for provision of lab space and permission of experiments, Susanne Busch and Regina Hansen for phytoplankton counting, and Joe Montoya for the provision of the Excel Sheet for calculations of uptake rates.

## Author Contributions

**Conceptualization:** Falk Eigemann, Maren Voss, Heide Schulz-Vogt.

**Data curation:** Falk Eigemann.

**Formal analysis:** Falk Eigemann, Angela Vogts, Luca Zoccarato.

**Funding acquisition:** Maren Voss, Heide Schulz-Vogt.

**Investigation:** Falk Eigemann, Angela Vogts, Maren Voss, Luca Zoccarato, Heide Schulz-Vogt.

**Methodology:** Falk Eigemann, Angela Vogts.

**Project administration:** Falk Eigemann.

**Resources:** Maren Voss, Heide Schulz-Vogt.

**Supervision:** Falk Eigemann, Maren Voss, Heide Schulz-Vogt.

**Validation:** Falk Eigemann.

**Visualization:** Falk Eigemann, Angela Vogts.

**Writing – original draft:** Falk Eigemann, Angela Vogts, Maren Voss, Luca Zoccarato, Heide Schulz-Vogt.

**Writing – review & editing:** Falk Eigemann.

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
