## [Decision Letter · Decision Letter 0]

24 Oct 2019

PONE-D-19-26135

Distinctive tasks of different cyanobacteria and associated bacteria in carbon as well as nitrogen fixation and cycling in a late stage Baltic Sea bloom

PLOS ONE

Dear Dr. Eigemann,

Thank you for submitting your manuscript to PLOS ONE. After careful consideration, we feel that it has merit but does not fully meet PLOS ONE’s publication criteria as it currently stands. Therefore, we invite you to submit a revised version of the manuscript that addresses the points raised during the review process.

Both reviewers found your manuscript to be of value, and only minor revisions were recommended by one reviewer. Please note and respond to the list of minor revisions listed in the attached reviewer's responses.

We would appreciate receiving your revised manuscript by Dec 08 2019 11:59PM. To enhance the reproducibility of your results, we recommend that if applicable you deposit your laboratory protocols in protocols.io, where a protocol can be assigned its own identifier (DOI) such that it can be cited independently in the future. For instructions see: http://journals.plos.org/plosone/s/submission-guidelines#loc-laboratory-protocols

We look forward to receiving your revised manuscript.

Kind regards,

O. Roger Anderson

Academic Editor

PLOS ONE

Journal Requirements:

Additional Editor Comments (if provided):

Reviewers' comments:

Reviewer's Responses to Questions

**Comments to the Author**

1. Is the manuscript technically sound, and do the data support the conclusions?

Reviewer #1: Yes

Reviewer #2: Yes

2. Has the statistical analysis been performed appropriately and rigorously? 

Reviewer #1: Yes

Reviewer #2: Yes

3. Have the authors made all data underlying the findings in their manuscript fully available?

Reviewer #1: Yes

Reviewer #2: Yes

4. Is the manuscript presented in an intelligible fashion and written in standard English?

Reviewer #1: Yes

Reviewer #2: Yes

5. Review Comments to the Author

Reviewer #1: I found this to a very interesting study with a major contribution by NanoSIMS analysis. The methods appear to be applied skillfully and the data presentation is clear and well illustrated. The article is generally very well-written in good English.

Reviewer #2: In this manuscript Eigmann et al. used Nano-SIMS to evaluate carbon and nitrogen fixation within a late-stage cyanobacterial bloom in the Baltic Sea. Surprisingly, they found considerable nitrogen fixation within heterotrophic bacteria and Pseudanabaena, and little to no fixation among heterocystous species.

Although the study is limited to a single sample (?) – time-series observations during bloom formation and collapse would have been very interesting – I found the results and analysis compelling. There are numerous small errors that need to be corrected before publication.

Citation styles are mixed, and the bibliography is incomplete

Line 32 – suggest “by NanoSims”

Line 37 – “but were also” is confusing, aren’t these the same thing?

Line 41 – “mass” is an odd term to use here, suggest bloom

Line 43-45 – sentence starting “The onset of” is nonsensical, please revise

Line 46 – at not “with”

Line 48 – up to not “for”

Line 51 – the mention of eukaryotic phytoplankton is a distraction, as they are not the subject of this study

Line 54 – here and elsewhere, “therewith” is not a word

Line 65 – suggest incapable in place of “not capable”

Line 66 – remove “and hence depend on dissolved nitrogen sources” as this is self-evident

Line 67 – I’m not an expert on these strains but a quick search for “Pseudanabaena nitrogen fixation” returned lots of hits with evidence of nitrogen fixation.

Line 70 – estuary not “estuarine”

Line 76 – delete “technique”

Line 84 – I’m not familiar with light traps, but as described it doesn’t make much sense; the zooplankton are removed but how does this result in concentration of cyanobacteria? What was the in situ chlorophyll concentration?

Line 106 – standard methods call for a 0.2 um filter for bacterial collection. Do you think you adequately captured the heterotrophic population?

Line 176 – “specific cell metabolism” requires some further explanation

Line 194 – where not “whereof”

Line 212 – I’m not following the logic of this statement; do you mean absolute numbers of cyanobacteria were not available?

Line 213 – the meaning of “2x1” is not clear

Line 214 – suggest “carbon content of … ratio of…”

Line 222 – cite R before R Studio

Line 233 – cite vegan

Table 1 – given the different units this table is a little hard to follow. I suggest using an asterisk or other symbol to identify those lines that correspond to heterotrophs.

Line 295-296 – these two sentences seem to be in opposition…

Line 323 – “Anova” should be capitalized

Line 341 – confirm that these a r, not r2

Line 352 – use time not “the time”

Line 409 – P-value should not be hyphenated

Line 444 – Comparable not “comparably”

Line 447 – do you mean one order of magnitude rather than one dimension?

Line 451 – suggest consistent with not “congenial”

6. PLOS authors have the option to publish the peer review history of their article (what does this mean?). If published, this will include your full peer review and any attached files.

Reviewer #1: No

Reviewer #2: Yes: Jeff Bowman

---

## [Author Response · Author response to Decision Letter 0]

18 Nov 2019

Response to reviewer

In this manuscript Eigmann et al. used Nano-SIMS to evaluate carbon and nitrogen fixation within a late-stage cyanobacterial bloom in the Baltic Sea. Surprisingly, they found considerable nitrogen fixation within heterotrophic bacteria and Pseudanabaena, and little to no fixation among heterocystous species.

Although the study is limited to a single sample (?) – time-series observations during bloom formation and collapse would have been very interesting – I found the results and analysis compelling. There are numerous small errors that need to be corrected before publication. 

Answer: We thank the reviewer for his profound comments and criticisms. The time series (10 min, 30 min, 1h, 6 h, 24 h) was set up in separate bottles, but the reviewer is right that they all originated from one location and one time point of the bloom. 

Citation styles are mixed, and the bibliography is incomplete 

Answer: We apologize for these mistakes. Please see the revised version for corrections.

Line 32 – suggest “by NanoSims”

Answer: Changed as suggested.

Line 37 – “but were also” is confusing, aren’t these the same thing?

Answer: The associated bacteria recycled nitrogen that was fixed by other organisms, but were also indicative of nitrogen fixation. Changed to:“… dominated the subsequent nitrogen remineralization with uptake rates up to 1.2 ± 1.93 fmol N h-1 cell -1, but were also indicative for fixation of di-nitrogen.”

Line 41 – “mass” is an odd term to use here, suggest bloom

Answer: Changed as suggested.

Line 43-45 – sentence starting “The onset of” is nonsensical, please revise

Answer: Changed to: “Start of blooms…“

Line 46 – at not “with”

Answer: Changed as suggested.

Line 48 – up to not “for”

Answer: Changed as suggested.

Line 51 – the mention of eukaryotic phytoplankton is a distraction, as they are not the subject of this study

Answer: Changed to: „Cyanobacteria live in close associations….“

Line 54 – here and elsewhere, “therewith” is not a word

Answer: Changed to „thereby“.

Line 65 – suggest incapable in place of “not capable”

Answer: Changed as suggested. 

Line 66 – remove “and hence depend on dissolved nitrogen sources” as this is self-evident

Answer: Removed as suggested.

Line 67 – I’m not an expert on these strains but a quick search for “Pseudanabaena nitrogen fixation” returned lots of hits with evidence of nitrogen fixation.

Answer: It is known for Pseudanabaena that it carries genes for nitrogen fixation. It was never shown, however, to perform nitrogen fixation in the Baltic Sea, despite several studies were performed on it. This might be related to the stage of the bloom, i.e. we sampled a late stage bloom, and other studies concentrated on active blooms.

Line 70 – estuary not “estuarine”

Answer: We apologize for this mistake. Changed as suggested.

Line 76 – delete “technique”

Answer: Deleted as suggested.

Line 84 – I’m not familiar with light traps, but as described it doesn’t make much sense; the zooplankton are removed but how does this result in concentration of cyanobacteria? What was the in situ chlorophyll concentration?

Answer: Changed to: „Positive phototactic zooplankton was removed by means of a light trap and bloom samples were concentrated until a cyanobacterial chl. a concentration…”

The original cyanobacterial chl. a concentration was ca. 4 µg l-1.

Line 106 – standard methods call for a 0.2 um filter for bacterial collection. Do you think you adequately captured the heterotrophic population?

Answer: We for sure did not capture the full heterotrophic population/diversity. This was, however, also not our goal. We were only interested in the directly to the cyanobacteria attached fraction of heterotrophic bacteria. After several pre-studies we decided to use 3-µm porewidth filter for this.

We added: „…pore width (we only aimed at the directly attached heterotrophic bacterial fraction) polycarbonate…”

Line 176 – “specific cell metabolism” requires some further explanation 

Answer: Changed to: „…of heterocysts in Aphanizomenon sp., Dolichospermum sp., and Nodularia sp. were avoided due to rapid transfer of fixed nitrogen.”

Line 194 – where not “whereof”

Answer: Corrected as suggested.

Line 212 – I’m not following the logic of this statement; do you mean absolute numbers of cyanobacteria were not available?

Answer: For cyanobacteria absolute numbers were available. Because of this, for cyanobacteria we could calculate uptake rates per volume per time. For the associated bacteria on the other hand, one can not calculate absolute numbers (they may be covered by organic material, attached to the backside of the host aso…). Because of this, we could only calculate uptake rates per cell for the associated bacteria.

For better understanding we changed this part to:

„For cyanobacteria gross uptake rates could be calculated per volume and time (absolute numbers were known). For the associated bacteria uptake rates were calculated per cell and time, because no absolute numbers of associated bacteria were existent.”

Line 213 – the meaning of “2x1” is not clear

Answer: Changed to: „…with 2 x1 (length x width) µm…”

Line 214 – suggest “carbon content of … ratio of…”

Answer: Changed as suggested.

Line 222 – cite R before R Studio

Answer: Citation changed as suggested.

Line 233 – cite vegan

Answer: Cited as suggested.

Table 1 – given the different units this table is a little hard to follow. I suggest using an asterisk or other symbol to identify those lines that correspond to heterotrophs.

Answer: Changed as suggested. Please see changes in the track changes version.

Line 295-296 – these two sentences seem to be in opposition…

Answer: Changed to: „For all time points, significant differences of 15N incorporation between the species/groups occurred (Fig. 6). After 30 min Pseudanabaena sp. (which reveals the highest 15N incorporation), and…”

Line 323 – “Anova” should be capitalized

Answer: Changed as suggested.

Line 341 – confirm that these a r, not r2

Answer: Yes, these are R values. R values are proxies for the degree of differences between different groups analysed by ANOSIM.

Line 352 – use time not “the time”

Answer: Changed as suggested.

Line 409 – P-value should not be hyphenated

Answer: Changed to: „…be adapted to this situation where phosphorus supply by degrading blooms may…”

Line 444 – Comparable not “comparably”

Answer: Changed as suggested.

Line 447 – do you mean one order of magnitude rather than one dimension?

Answer: Changed to: „…i.e. approximately one order of magnitude above…”

Line 451 – suggest consistent with not “congenial” 

Answer: Changed as suggested.

---

## [Editor Report · Decision Letter 1]

20 Nov 2019

Distinctive tasks of different cyanobacteria and associated bacteria in carbon as well as nitrogen fixation and cycling in a late stage Baltic Sea bloom

PONE-D-19-26135R1

Dear Dr. Eigemann,

We are pleased to inform you that your manuscript has been judged scientifically suitable for publication and will be formally accepted for publication once it complies with all outstanding technical requirements.

With kind regards,

O. Roger Anderson

Academic Editor

PLOS ONE

Additional Editor Comments (optional):

Thank you for submitting your revised manuscript and carefully addressing the minor recommendations of the two reviewers.
---

## [Editor Report · Acceptance letter]

6 Dec 2019

PONE-D-19-26135R1 

Distinctive tasks of different cyanobacteria and associated bacteria in carbon as well as nitrogen fixation and cycling in a late stage Baltic Sea bloom 

Dear Dr. Eigemann:

I am pleased to inform you that your manuscript has been deemed suitable for publication in PLOS ONE. Congratulations! Your manuscript is now with our production department. 

With kind regards,

on behalf of

Dr. O. Roger Anderson 

Academic Editor

PLOS ONE